# Cryo-EM structure of a licensed DNA replication origin

Ferdos Abid Ali[1], Max E. Douglas[2], Julia Locke[1], Valerie E. Pye ⬤ [3], Andrea Nans[4], John F.X. Diffley ⬤ [2] & Alessandro Costa[1]

Eukaryotic origins of replication are licensed upon loading of the MCM helicase motor onto DNA. ATP hydrolysis by MCM is required for loading and the post-catalytic MCM is an inactive double hexamer that encircles duplex DNA. Origin firing depends on MCM engagement of Cdc45 and GINS to form the CMG holo-helicase. CMG assembly requires several steps including MCM phosphorylation by DDK. To understand origin activation, here we have determined the cryo-EM structures of DNA-bound MCM, either unmodified or phosphorylated, and visualize a phospho-dependent MCM element likely important for Cdc45 recruitment. MCM pore loops touch both the Watson and Crick strands, constraining duplex DNA in a bent configuration. By comparing our new MCM–DNA structure with the structure of CMG–DNA, we suggest how the conformational transition from the loaded, post-catalytic MCM to CMG might promote DNA untwisting and melting at the onset of replication.

[1] Macromolecular Machines Laboratory, The Francis Crick Institute, 1 Midland Road, London NW1 1AT, UK. [2] Chromosome Replication Laboratory, The Francis Crick Institute, 1 Midland Road, London NW1 1AT, UK. [3] Chromatin Structure and Mobile DNA Laboratory, The Francis Crick Institute, 1 Midland Road, London NW1 1AT, UK. [4] Structural Biology of Cells and Viruses, The Francis Crick Institute, 1 Midland Road, London NW1 1AT, UK. Correspondence and requests for materials should be addressed to A.C. (email: alessandro.costa@crick.ac.uk)

Replisomes in all cellular systems contain a helicase that unwinds DNA, providing the template for chromosome duplication[1]. In eukaryotes, helicase recruitment at replication origins and DNA melting are temporally separated. The loading of two hexameric Minichromosome-Maintenance (MCM) motors onto DNA occurs during the G1 phase of the cell cycle and involves the formation of an inactive MCM double hexameric ring that encircles the double helix[2,3]. The switch into S phase requires a cascade of events, including MCM phosphorylation by Dbf4-dependent kinase (DDK)[4,5]. This phosphorylation generates a binding site for Sld3/7 which in turn promotes the recruitment of helicase activator, Cell-division-cycle 45 (Cdc45)[6]. Subsequently Go-Ichi-Ni-San (GINS) becomes associated with the MCM, in a process that requires additional phosphorylation by Cyclin-dependent kinase (CDK) along with other firing factors including DNA polymerase B 11 (Dpb11), Synthetically lethal with Dpb11 2 (Sld2) and DNA polymerase $\varepsilon$[7,8]. The Cdc45–MCM–GINS (CMG) complex is the active form of the helicase[9,10] and is believed to (i) melt the double helix, (ii) eject one strand from the ring pore (possibly aided by additional firing factors), and (iii) unwind the DNA at the replication fork, by working as a processive single-stranded DNA translocase[11]. Elucidating origin melting at the start of DNA replication requires an understanding of the molecular basis of DNA engagement by the MCM and the mechanics of the helicase motor. The MCM belongs to the superfamily of ATPases associated with various cellular activities (AAA+) ATPases, which bind and hydrolyze ATP at inter-subunit active sites[12] to catalyze DNA loading and double hexamer formation during origin licensing first[13,14], and then to open and unwind DNA upon origin firing[15]. Multiple MCM pore loops have been implicated in DNA binding, including the N-terminal B-domain zinc fingers, OB (oligosaccharide/oligonucleotide) hairpins and the ATPase motor hairpins, PS1 (Presensor 1) and h2i (helix-2 insertion)[16]. Whether the structure of MCM is altered upon DDK phosphorylation is unclear and how MCM enables DNA melting is not understood. To address these questions, we have started to analyze the reconstituted DNA replication system with purified yeast proteins by single-particle cryo-electron microscopy (EM)[2,17].

We report here the structure of the DNA-loaded MCM double hexamer before and after DDK phosphorylation. Previously unreported N-terminal MCM elements (probably the targets of DDK) become visible upon phosphorylation, while the configuration of the MCM core particle remains unchanged in the unmodified and phosphorylated forms. Duplex DNA threads through the entire length of the double hexamer and multiple MCM pore elements contact the double helix, interacting with both the Watson and Crick strands. Comparative analysis of our MCM–DNA structure and a structure of CMG–DNA[18] highlights conformational changes in both the N-terminal tier and ATPase motor domain of MCM, suggesting a model for how DNA can become untwisted during origin firing. Combined with other recent structures of MCM and its interactors[18–25], our results provide important insights into origin licensing and activation in eukaryotic cells.

## Results

### Effect of DDK phosphorylation on the MCM double hexamer.
MCM phosphorylation by DDK is required for Cdc45 recruitment mediated by the Sld3/7 complex[6]. While it is known that phosphorylation does not affect double-hexamer stability, it is unclear whether it causes any structural rearrangements in the MCM core[26]. The N-terminal tails of the adjacent helicase subunits Mcm4 and Mcm6 are targeted by DDK and

phosphorylation promotes Sld3 recruitment to the MCM double hexamer[6]. The N-terminal region of Mcm4–6 is therefore likely to constitute a landing pad for Sld3/7, however this element has so far eluded structural characterization. To investigate the effect of DDK phosphorylation on the double hexamer, we imaged the DNA-bound MCM before and after DDK phosphorylation. To this end, MCM particles were loaded onto bead-immobilized DNA, building on a previously established protocol (Fig. 1a)[2]. Loaded MCM was efficiently phosphorylated by DDK[6,26], as evidenced by the change in MCM-band mobility shown in Fig. 1b. After a high-salt wash to remove the MCM loading factors and DDK, DNA was released from the beads by restriction enzyme digest (Fig. 1b) and applied to a lacey continuous-carbon grid for high-resolution cryo-EM imaging (Supplementary Fig. 1). Inspection of the derived two-dimensional (2D) classes (Fig. 1c) of the unphosphorylated complex reveals the recognizable four-tier configuration of MCM double hexamers (comprising two N-terminal dimerization tiers sandwiched by two ATPase tiers, Fig. 1c)[2,24]. DDK-phosphorylated particles present additional density symmetrically decorating the dimerization domain, best appreciated upon inspection of a difference image (Fig. 1c). As DDK is washed away before elution of the double hexamer, this new feature must belong to MCM. To map this element on the double hexamer (Fig. 1c), we determined three-dimensional structures from both unmodified and DDK-phosphorylated particles. Following two- and three-dimensional classification, the structures of the unmodified and phosphorylated MCM were refined to 4.9 Å and 4.65 Å resolution (before map sharpening), respectively (Supplementary Figs. 1–4). In both structures, the two MCM rings are slightly misaligned (Fig. 1d)[24], forming a continuous central pore that narrows around the N-terminal dimerization interface (Fig. 1e). While no significant conformational change is detected upon phosphorylation in the core MCM particle (Supplementary Movie 1), in the structure of the DDK-treated MCM visualized at lower contour level, additional density wraps around Mcm4 and 6, sitting across the MCM N-terminal dimerization interface (Fig. 1f). Although clearly visible in a 3D difference map, the new density is poorly resolved, suggesting that it might represent the average of distinct conformers. Local 3D classification allowed us to identify two states for this density feature ("extended" and "compact") and two-dimensional classification of the derived particle subsets confirms the existence of two conformations (Supplementary Fig. 5 and Supplementary Movie 2). We failed to isolate a subset of DDK-treated particles resembling unphosphorylated MCMs, or a fraction of unphosphorylated particles containing the new N-terminal element. Given these results, our preferred interpretation is that new flexible element corresponds to the N-terminal tails of Mcm4 and 6, which become visible upon DDK phosphorylation. This phospho-dependent element likely becomes more rigid, although still flexibly tethered to the core of the MCM complex, resulting in detectable extra density. Phospho-Mcm4,6 would then serve as a landing platform for the recruitment of Sld3/7, and the deposition of Cdc45 onto the MCM[15] (Fig. 1g).

### The ATPase centers in the reconstituted MCM double hexamer.
ATP hydrolysis by MCM is essential for double hexamer formation[13,14]. To understand nucleotide binding and release upon MCM loading, we sought to determine the structure of the double hexamer at higher resolution. After map sharpening, the unmodified and phosphorylated MCM structures were visualized at 4.8 Å and 4.3 Å resolution respectively (Supplementary Figs. 1–4), and were used to refine an atomic model (Fig. 2a and Supplementary Fig. 6). Although at these resolutions ATP cannot be distinguished from ADP with confidence, nucleotide occupancy of

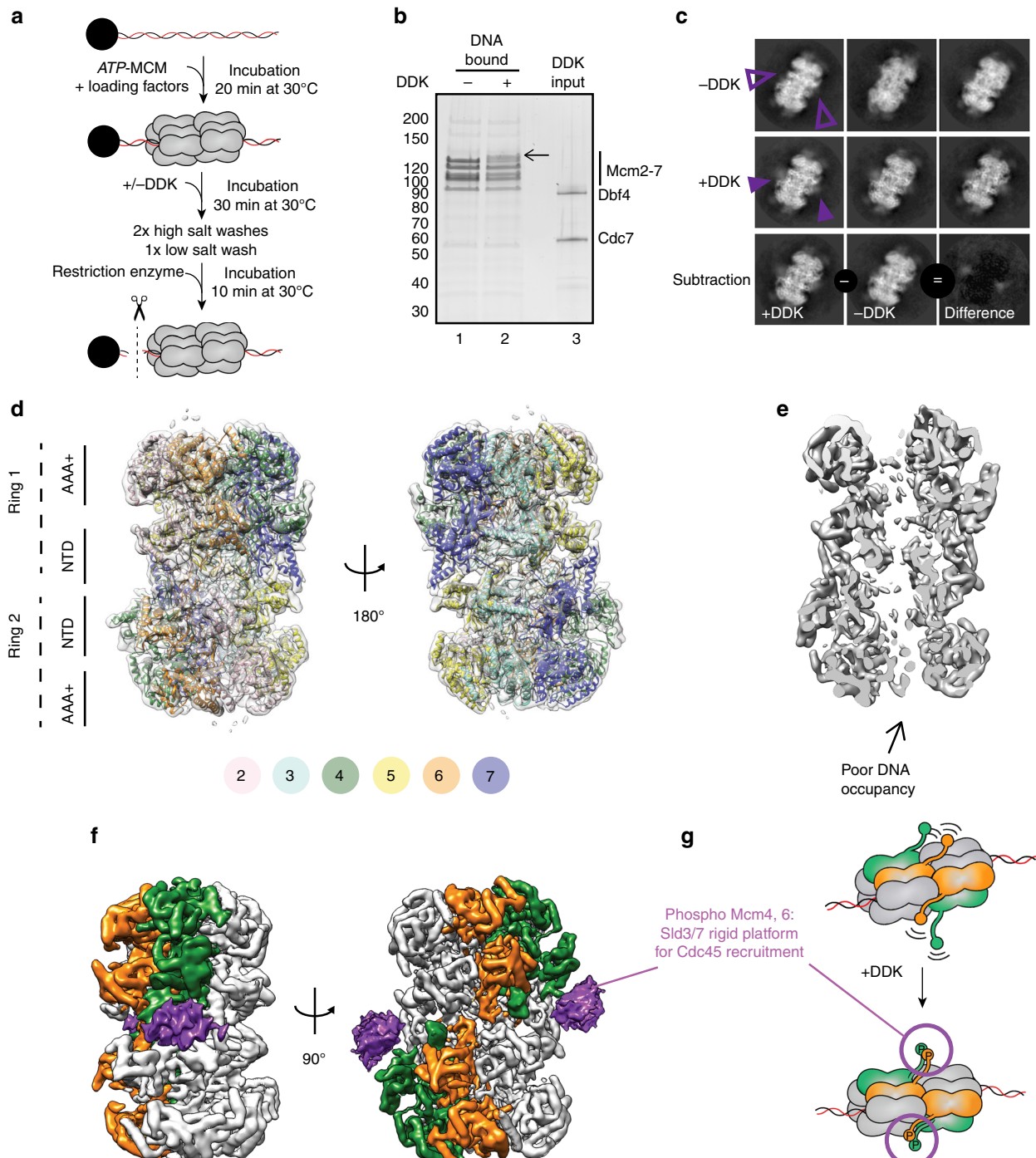

**Fig. 1** Structure of the unmodified and DDK-phosphorylated MCM double hexamer. **a** Scheme for the loading and phosphorylation of the MCM double hexamer on biotinylated duplex DNA. DDK was removed by high-salt wash. MCM particles retained on digested DNA were used for cryo-EM experiments. **b** Silver-stained SDS–PAGE gel of (lane 1) unmodified and (lane 2) phosphorylated MCM compared to the DDK input (lane 3). No trace of DDK is found in the MCM lanes. A black arrow points at the shifted phospho-Mcm6. **c** 2D class averages of the unmodified (first row) and DDK phosphorylated MCM double hexamer (second row). The third row contains a difference image obtained by subtracting unphosphorylated from phosphorylated MCM. Purple arrowheads point to the absence (hollow) or presence (filled) of additional density surrounding the double-hexamer interface in the unmodified and DDK phosphorylated samples, respectively. **d** 4.65 Å resolution (unsharpened map) of the DDK phosphorylated MCM double-hexamer with docked in atomic coordinates of yeast MCM (PDB: 3ja8). **e** A cut-through view of the DDK phosphorylated MCM double hexamer (unsharpened map) reveals a narrowing of the channel at the N-terminal domain interface and a poor occupancy of DNA. **f** Phosphorylated MCM displayed at σ = 0.37, superposed to a difference map displayed at σ = 0.08 (purple). To obtain the difference map the unmodified MCM was subtracted from the DDK phosphorylated MCM. **g** Cartoon depicting the effect of MCM phosphorylation by DDK. Phosphorylation increases rigidity to N-terminal Mcm4-6, which become visible upon averaging

the ATPase centers can clearly be appreciated (Fig. 2b). ATPase ring inspection reveals that only Mcm5-3 and Mcm2–5 are tightly nucleotide-bound. Because ATP hydrolysis by Mcm2–5 and Mcm5-3 is required for MCM loading onto DNA[13], we assign these nucleotide densities to ADP. The remaining ATP hydrolysis centers contain much weaker nucleotide density and the Mcm6-2 site is found in a relaxed configuration likely reflecting nucleotide release. The same Mcm5-3 and Mcm2–5 hydrolysis centers are found to be ATP bound in the yeast CMG–DNA structure[18] (obtained in the presence of ATPγS, Fig. 2b). Notably, the published endogenous MCM double hexamer structure presents a different ATPase occupancy profile, with nucleotide populating all catalytic sites[24]. This difference is likely because the endogenous MCM was imaged in a buffer containing 3 mM ATP, while excess nucleotide was washed off in our reconstituted system.

**Cryo-EM structure of the MCM double hexamer on DNA**. Understanding helicase loading requires a reliable MCM–DNA structure but only a fraction of the MCM particles are DNA-associated in our data set, probably due to double-hexamer sliding[2], resulting in poor DNA occupancy in the ensemble structure, similar to a previous cryo-EM report[24] (Fig. 1e). To obtain a better nucleoprotein structure we sought to identify a subset of duplex-DNA engaged MCM. However, apo and DNA-bound double hexamers cannot easily be discriminated using 2D classification (Supplementary Fig. 1), due to the preferential orientations perpendicular to the ring channel. Similarly, 3D classification attempts to retrieve a fully occupied DNA-MCM structure have failed. To circumvent this problem, we developed a protocol to "peek" inside the ATPase ring using 2D analysis, based on available signal subtraction methods[27] (Fig. 3a). We generated a mask for the ATPase domain from the ensemble 3D structure and subtracted the masked density from the aligned images at the raw particle level (Fig. 3a). Subsequent 2D classification of the ATPase-subtracted MCM particles yielded a number of classes with recognizable N-terminal domain dimers but lacking the ATPase domains. While most 2D averages revealed no DNA occupancy, a subset of classes (20% for unmodified, 7% for DDK phosphorylated MCM) showed a recognizable helical DNA density coaxial with the ring central channel and symmetrically departing from the MCM dimerization domain (Fig. 3b). No evidence was found for DNA departing from the N-terminal MCM at different angles[28], supporting the notion that DNA spans the length of the entire MCM double hexamer. To establish whether DNA maintains its double-helical nature through the MCM dimerization interface, we selected all particles contributing to the ATPase-subtracted MCM–DNA class averages and reverted back to the original images, to reconstruct a full MCM–DNA structure. This approach yielded a resolution of 6.9 Å for the unmodified and 7.5 Å resolution for the phosphorylated MCM structure (Supplementary Figs. 7–9), showing continuous DNA density that runs through the central pore of the entire MCM double hexamer (Fig. 3c). These structures were obtained using two-fold symmetry averaging, which could in principle bias DNA arrangement across the two hexamers. To exclude this possibility, we have solved an ensemble structure of MCM–DNA in the absence of any symmetry imposition to a resolution of 7.2 Å, showing virtually identical DNA density (Supplementary Fig. 10).

The resolution achieved allows for the discrimination of major and minor grooves in the DNA, and for the generation of a model of the MCM-bound DNA double helix. In total 62 base pairs of DNA were built into the DNA density, in agreement with previous estimations[2] (Fig. 4a, b). DNA becomes bent by 10° as it

enters the dimerization domain and maintains its double helical nature throughout the MCM channel (Fig. 4b). Rigid-body docking of our DNA-free double hexamer into the DNA-bound structure resulted in an excellent fit, allowing us to generate a composite-resolution model for the DNA–MCM structure (Fig. 4c).

Both the Watson and Crick strands in the double helix are contacted by seven distinct pore loop elements in each MCM ring. These include the zinc fingers of Mcm2,5,6, the OB hairpins of Mcm4,7, the h2i hairpin of Mcm2 and the PS1 hairpin of Mcm6 (Watson) and the OB hairpins of Mcm2,3,5, and the h2i hairpins of Mcm3,4,5,7 (Crick) (Fig. 4d, Supplementary Fig. 11). In conclusion, several MCM pore-loop interactions engage DNA (although a short-lived, disengaged intermediate probably exists, which allows sliding). The observation that each MCM in the double hexamer binds both DNA filaments likely serves as a measure to prevent directional translocation before DNA replication initiation.

**A model for CMG-mediated DNA melting**. The initial unwinding of origin DNA during replication initiation must take place after assembly of the MCM double hexamer, but this process is poorly understood at a molecular level[11]. Our structure of the yeast MCM double hexamer bound to duplex DNA can be compared to the structure of the yeast CMG engaged with a replication fork[18]. This exercise suggests a model for how the conformational changes in MCM that are induced by CMG assembly may promote DNA melting at the onset of replication. To model this transition, one MCM ring bound to duplex DNA was extracted from the double hexamer and the MCM bound to forked DNA was extracted from the CMG[18]. The two resulting MCM rings were aligned via their N-terminal domains. By morphing between the inactive and the active helicase forms, two large-scale reconfigurations of the protein component can be observed. The ATPase tier, which is originally slightly offset in the complex, translates and becomes aligned with the N-terminal hexamer axis, while both N-terminal and ATPase domains tighten around DNA (Supplementary Movie 3). The double helix exiting from the N-terminal domain transitions from intimately interacting with the Mcm2 and Mcm5 side of the ring, to engaging with the Mcm4 and Mcm7 side. In this transition, DNA rotates clockwise, favouring duplex DNA underwinding (Fig. 5a, b). In the CMG–forked–DNA complex, incoming duplex DNA is clearly visible, the excluded-strand cannot be observed due to flexibility and the ATPase-captured strand is well-resolved[18]. The MCM-captured single-stranded DNA filament becomes stretched and untwisted as it traverses the AAA+ tier. Inspection of the pore loop elements that contact DNA in both the inactive and active MCM allow us to propose a model for DNA stretching. According to our proposed mechanism, transition from the inactive (Fig. 5c) to the active helicase causes the movement of a number of pore loops, which maintain DNA engagement (Fig. 5d). Mcm4 OB hairpin remains fixed, mapping close to the duplex/single-stranded DNA junction in the CMG. Mcm7 OB hairpin moves downwards by 9 Å and pushes against the DNA junction, while PS1 hairpin of Mcm6 moves upwards by 12 Å (Supplementary Movie 3). We note that these two counteracting movements would result in DNA stretching and melting, likely promoting duplex DNA untwisting. Concomitantly, Mcm3 and Mcm5 h2i hairpins would become poised to stabilize the stretched MCM-captured strand in the CMG (Fig. 5c, d, Supplementary Movie 3). In summary, we propose a model whereby the counteracting movements of Mcm6 and Mcm4/7, combined with the single-stranded DNA stabilizing interactions of Mcm3 and Mcm5 result in stretching and untwisting of DNA.

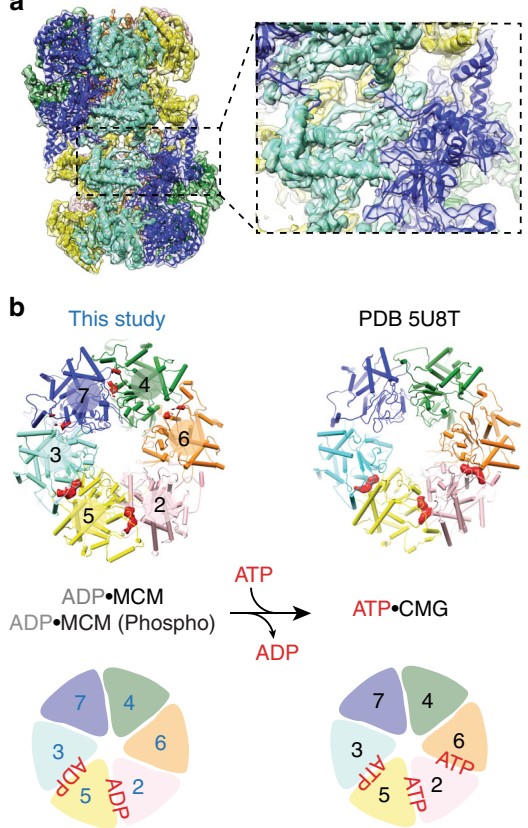

**Fig. 2** Near atomic resolution model of MCM and nucleotide occupancy of the ATPase motor. **a** 4.3 Å resolution structure of the phosphorylated MCM and a detail of the dimerization interface shown on the right. **b** On the left: atomic model of the MCM ATPase motor, overlaid with the nucleotide density recovered in between AAA+ protomers. Mcm5-3 and Mcm5-2 contain strong nucleotide density peaks. The same ATPase site profile can be observed for the unphosphorylated form of MCM. Mcm2-6 is nucleotide free and in a relaxed configuration favouring ADP release. On the right: atomic model of ATP-CMG. Mcm5-3, Mcm2-5 and Mcm6-2 are bound to ATP

Concomitantly, duplex DNA rotates upon helicase activation, promoting DNA unwinding (Supplementary Movie 3). Our model for DNA melting assumes that MCM double hexamer to CMG transition occurs on-path, however CMG formation requires intermediate helicase maturation steps, including DDK —dependent Sld3/7/Cdc45 recruitment and CDK-dependent Sld2-Dpb11-Pol epsilon-GINS recruitment. Our DNA untwisting mechanism does not model the melted strand, because it is invisible in the CMG–DNA structure[18]. We note however that, upon DNA rotation, this strand would become poised to interact with the so-called "MCM single-stranded Binding" (MSSB) site, a melted-DNA capture element mapping in the N-terminal region of Mcm467[29]. However, our results do not address which strand inside the MCM channel eventually serve the function of translocation substrates at the replication fork[30]. Biochemical and structural analysis of replisome maturation, DNA melting and replication fork establishment will be key to understand origin activation.

## Discussion

Origin licensing entails the loading of an inactive double MCM hexamer onto duplex DNA[2,3]. Subsequently, replication initiation requires opening of the double helix and the MCM-engagement of multiple replisome components, including Cdc45/GINS[1,17]. These three steps precede replication fork establishment in eukaryotic cells. The study presented here increases our understanding of key stages in origin licensing, replisome assembly and origin DNA melting. In our MCM–DNA structures the double helix spans the entire length of the double hexamer (Fig. 3c). This observation argues against recently proposed models suggesting that origin licensing alone might promote DNA melting at the MCM dimerization interface[24,31]. We also see no evidence that DNA could thread through one MCM ring and egress through a side gate in the second ring[28]. In each of the two hexamers, we find that MCM pore loops provide 14 distinct contact points with the double helix and equally engage the Watson and Crick strands (Fig. 4a, d). This observation is surprising because hexameric ATPases that move on duplex DNA generally track on one strand only[32]. Our finding provides a rationale for why the MCM motor can equally bind a polyT forked-DNA substrate in vitro, in either a productive and non-productive fashion[33] (i.e.,

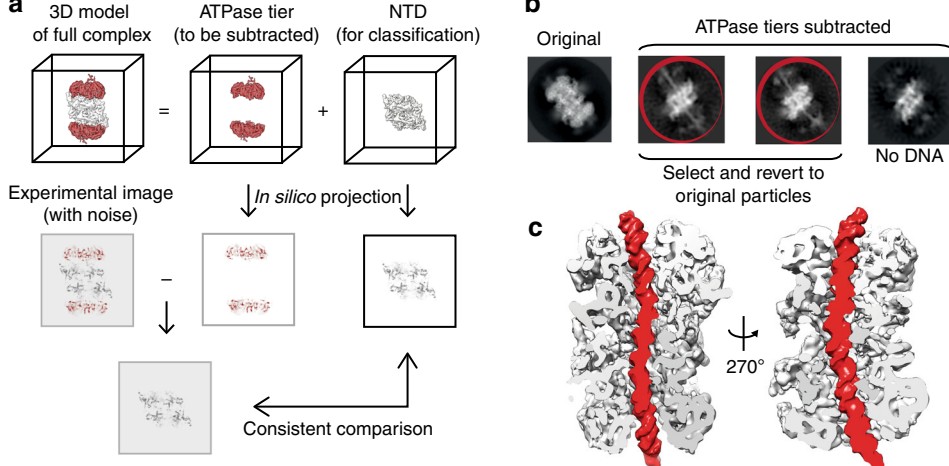

**Fig. 3** Signal subtraction of ATPase motor allows structural characterization of the DNA-bound double hexamer. **a** Signal subtraction scheme, based on Bai et al.[27]. The full 3D model is composed of the AAA+ ATPase and N-terminal tiers. A mask was created for the ATPase tier, which was applied to the experimental images (particles) for subtraction. The remaining density corresponds to N-terminal tiers and DNA (if present) that would emanate from the channel. **b** Signal subtracted particles were subjected to 2D classification. Classes showing helical DNA density protruding from the N-terminal tiers were selected and reverted to their original full density before 3D refinement. **c** Cryo-EM map of the DNA-bound, unphosphorylated MCM double hexamer

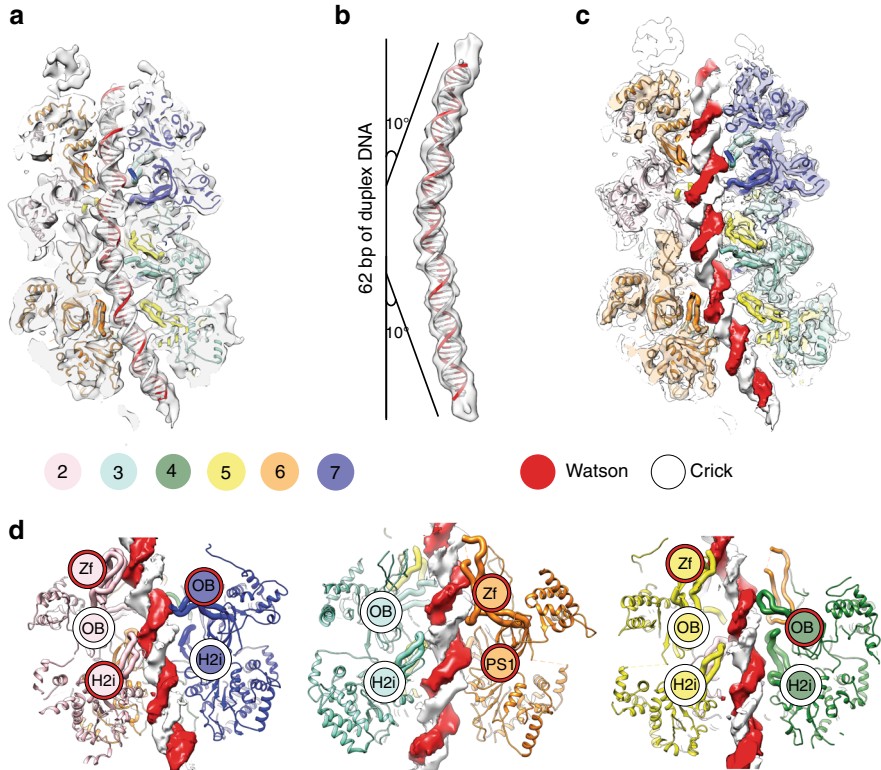

**Fig. 4** Structure of the MCM double hexamer bound to duplex DNA. **a** Cut-through view of the cryo-EM map (6.9 Å resolution) with the refined MCM atomic coordinates. Both Watson and Crick strands are touched by MCM pore loops in each single hexamer. **b** 62 base pairs of duplex DNA were built inside the MCM channel. The double helix bends by 10° as it enters the N-terminal dimerization domain. **c** Overlay between the 6.9 Å resolution, DNA-occupied structure and the higher-resolution ensemble protein structure fails to highlight detectable conformational changes in MCM upon DNA binding. A composite resolution structure can therefore be built integrating the atomic models derived from the lower resolution DNA and higher-resolution protein structures. **d** Detailed analysis of the MCM pore loops engaging the Watson and Crick strands. OB, oligosaccharide/oligonucleotide fold. PS1, Pre-sensor 1 hairpin. Zf zinc-finger hairpin, H2i helix-2-insert

with either the Watson strand or Crick strand trapped inside the MCM channel). Also, promiscuous strand engagement explains how opposite polarity of fork binding have been reported, depending on the model system used and the experimental conditions[18,34–36]. A similar duplex DNA-binding mode, with AAA+ hexamer elements touching both strands, has recently been reported for SV40 Large T antigen, a replicative helicase that shares multiple mechanistic aspects with MCM, including ring dimerization at a replication origin[37]. We suggest that promiscuous DNA strand engagement in the helicase double hexamer might represent a universal mode of inhibiting directional movement on the DNA prior to replisome maturation.

A second finding of our study can be derived from the inspection of the ATPase sites. The resolution obtained for our structures allows us to detect nucleotide occupancy in the six ATPase sites and reveals that Mcm2–5 and Mcm5–3 maintain a tight nucleotide interaction that tolerates extensive washes in nucleotide-free buffer. Conversely, Mcm6–2 is found in an open configuration, which likely favors nucleotide release (Fig. 2b). Double hexamer formation requires ATP hydrolysis in both Mcm2–5 and Mcm5–3, suggesting that the loaded MCM might be ADP bound. We note that the CMG–DNA structure contains ATP in the Mcm2–5, Mcm5–3, and Mcm6–2 active sites[18]. It remains to be established whether CMG formation actually promotes nucleotide exchange at these sites, which are indeed known to play a major role in DNA unwinding[10]. The remaining Mcm4–6, 7–4, and 3–7 sites are instead dispensable for helicase activity[10], but are important for double hexamer formation[13,14]. Whereas in our double hexamer structure Mcm4–6,7–4,3–7 are

nucleotide-free (Fig. 2b), the same sites are occupied in an earlier loading intermediate[20], indicating that different ATPase sub-complexes in the MCM have evolved specialized roles during origin licensing and activation[31].

The first step towards CMG formation is the phosphorylation of the Mcm4 and Mcm6 N-terminal tails, necessary for the recruitment of the Sld3/7 complex that chaperones Cdc45 onto the MCM[5,6]. In comparing the subnanometre resolution structure of the unmodified and phosphorylated MCM, we could not observe any significant structural changes in the MCM core particle (Supplementary Movie 1). However, previously unreported electron density symmetrically decorating the N-terminal dimerization interface of MCM could be detected at lower contour level, protruding from Mcm4 and 6 (Fig. 1f). Because DDK is washed off MCM during our protocol, we propose that the new density feature corresponds to the phosphorylated N-terminal tails of Mcm4 and 6, which become relatively more stable, compared to their unmodified counterparts (hence visible upon averaging, Fig. 6). Future work will establish whether these two-fold symmetric features serve as a landing platform for the Sld3/Sld7 dimer of dimers, which could reveal a symmetric mechanism of Cdc45 recruitment[38].

Double hexamer MCM formation is not sufficient to melt DNA and opening of the double helix must occur during a following step[11]. Our new structural data suggests how origin DNA melting might occur. To understand duplex DNA untwisting, we have compared our new MCM–DNA structure and the previously published atomic model of yeast CMG bound to DNA[18]. Upon CMG formation, the DNA translates and rotates,

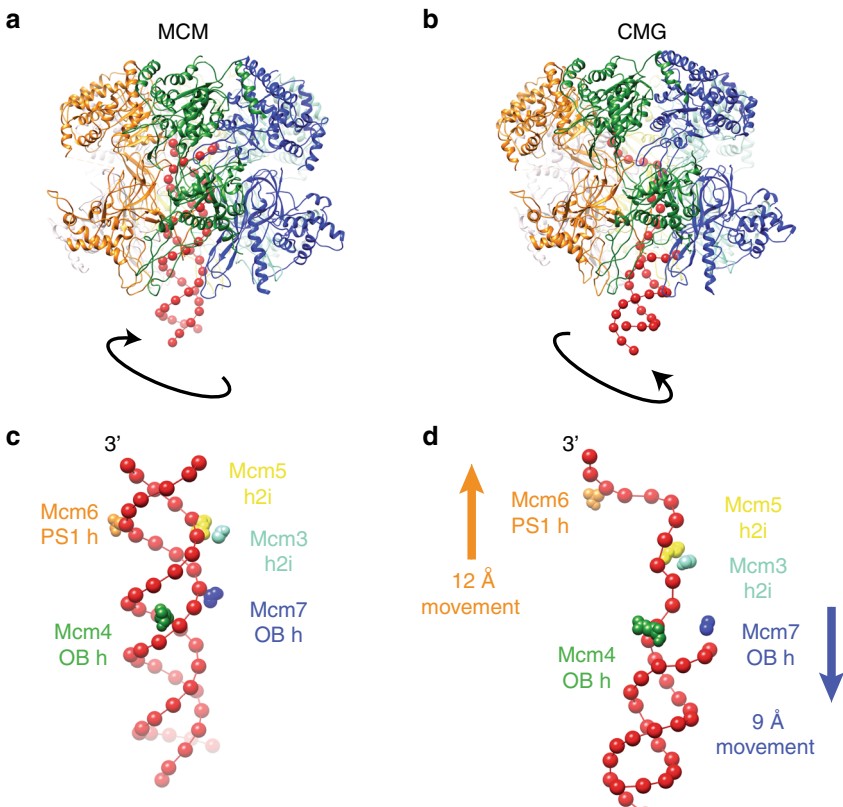

**Fig. 5** A model for origin DNA opening: MCM-to-CMG transition causes DNA melting inside the helicase channel. **a**, **b** Duplex DNA encircled by the N-terminal MCM Zinc fingers becomes underwound as it transitions from MCM to CMG. Concomitantly, the ATPase-captured strand becomes stretched and untwisted inside the helicase pore. **c** MCM pore loop elements interacting with duplex DNA. Notably, ATPase pore loops Mcm6 PS1 h and Mcm3/5 h2i hairpins engage both the Watson and Crick strands. **d** MCM pore loops in the CMG touch the stretched and untwisted DNA. Mcm6 PS1 hairpin moves upwards by 12 Å. Mcm7 OB hairpin moves downwards by 9 Å and pushes against the DNA junction. Mcm4 remains fixed in proximity to the DNA junction. Mcm3/5 h2i hairpins stabilize the stretched configuration of the ATPase-captured strand

promoting unwinding (Fig. 5a, b). Pore loop elements that contact DNA in both structures move apart from one another and hence stretch the Watson strand, trapped between the N-terminal and ATPase tiers of MCM (Fig. 6). *Drosophila melanogaster* CMG binding to single-stranded DNA also promotes DNA stretching, as we have previously shown using single-molecule FRET[25]. This supports the notion that the mechanism of DNA deformation by the CMG is likely to be conserved from yeast to metazoa. We derive a mechanism whereby stretching of one DNA filament destabilizes the double helix, promoting DNA untwisting and origin melting. Certain mechanistic aspects of origin DNA opening appear to be conserved across various domains of life. A previous study combining crystallography and FRET-based assays, in fact, has demonstrated a similar mechanism for DnaA, the AAA + ATPase that opens origin DNA in bacteria[39]. DnaA is not a helicase but rather a replication initiator that partakes in replicative helicase recruitment. Therefore, although both bacteria and eukarya are likely to melt origin DNA by stretching one strand to untwist the double helix, they do so with diverse replication components[1]. Importantly, while DNA wrapping by DnaA contributes to duplex-DNA melting during replication initiation in bacteria[1], a similar function is not required for eukaryotic origin activation[17].

The mechanism of replication fork establishment still remains elusive. In particular, our model does not explain what happens to the melted DNA upon origin opening. We note however that the Mcm467 side of the helicase ring appears available for DNA binding and could trap this strand upon origin melting. Mcm467

contains the N-terminal MSSB melted-DNA-capture element identified in archaea and yeast[29], as well as a second single-stranded-DNA interaction site observed in the cryo-EM structure of *Drosophila* CMG[25]. Whether the Watson or the Crick strand serves as the translocation substrate for the CMG and how the excluded strand is ejected from the MCM ring for replication fork establishment remain debated issues[18,34–36]. Our future efforts will be devoted towards understanding replication fork movement using the yeast DNA replication reconstituted system imaged by cryo-EM.

## Methods

**Protein purification.** Cdt1–Mcm2–7 was purified from budding yeast strain yAM33 arrested in G1 phase for 3 h with alpha factor[40]. Expression was induced at 30 °C by addition of galactose to 2%, and continued for 3 h before cells were collected. Cells were lysed in 25 mM Hepes pH 7.6, 5 mM MgOAc, 100 mM K-glutamate, 0.02% NP-40 substitute (NP-40S) and 10% glycerol (buffer MCM) supplemented with protease inhibitor tablets (Roche) by freezer milling. An equal volume of buffer MCM was added to cell powder thawed at room temperature, the lysate was cleared by centrifugation for 1 h at 235,000 g, 4 °C and the supernatant supplemented with 2 mM CaCl₂. Cdt1–Mcm2–7 complex was affinity purified with calmodulin resin (Roche), eluted using 5 mM EDTA, and protein containing fractions concentrated and loaded onto a superdex 200 gel filtration column equilibrated in buffer MCM. Fractions containing heptameric complex were pooled, concentrated and snap frozen in liquid nitrogen for storage.

ORC was purified from budding yeast strain ySDORC[40] arrested in G1 phase for 3 h with alpha factor. Expression was induced at 30 °C by addition of galactose to 2%, and continued for 3 h before cells were collected. Cells were lysed in 25 mM Hepes pH 7.6, 300 mM KCl, 0.05% NP-40 substitute (NP-40S) and 10% glycerol (buffer ORC) supplemented with protease inhibitor tablets (Roche) by freezer milling. An equal volume of buffer MCM was added to cell powder thawed at room temperature, the lysate was cleared by centrifugation for 1 h at 235,000×g, 4 °C and

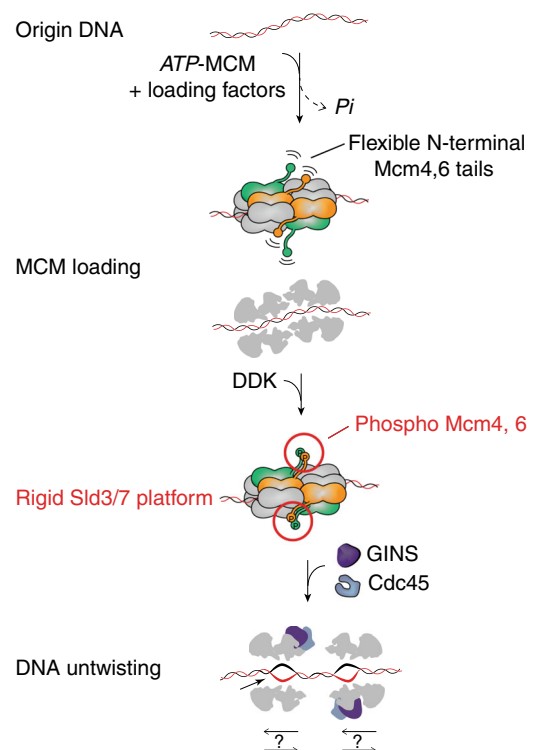

**Origin DNA**

*ATP*-MCM
+ loading factors → *Pi*

Flexible N-terminal
Mcm4,6 tails

**MCM loading**

DDK

Phospho Mcm4, 6

Rigid Sld3/7 platform

GINS
Cdc45

**DNA untwisting**

?   ?

**Fig. 6** Model for replication initiation. MCM is loaded onto origin DNA in a process that requires ATP hydrolysis by MCM. In the loaded double hexamer, duplex DNA runs through the entire length of the central channel. Upon DDK phosphorylation the N-terminal tails of Mcm4-6 become more stable, probably forming a landing platform for the Cdc45-recruiting factor, Sld3/Sld7. Upon binding of GINS and Cdc45, MCM melts the double helix. The directionality of CMG movement after replication fork establishment remains a matter of debate

the supernatant supplemented with 2 mM CaCl$_2$. ORC complex was affinity purified with calmodulin resin (Roche), eluted using 5 mM EDTA, and protein containing fractions concentrated and loaded onto a superdex 200 gel filtration column equilibrated in buffer ORC. Fractions containing ORC complex were pooled, concentrated, dialysed against buffer ORC with the KCl concentration reduced to 200 mM and snap frozen in liquid nitrogen for storage.

DDK was purified from budding yeast strain ySDK8[26]. Expression was induced at 30 °C by addition of galactose to 2%, and continued for 5–8 h before cells were collected. Cells were lysed in 25 mM Hepes pH 7.6, 400 mM NaCl, 0.05% NP-40 substitute (NP-40S) and 10% glycerol (buffer DDK) supplemented with protease inhibitor tablets (Roche) by freezer milling. An equal volume of buffer DDK was added to cell powder thawed at room temperature, the lysate cleared by centrifugation for 1 h at 235,000 g, 4 °C and the supernatant supplemented with 2 mM CaCl$_2$. DDK was affinity purified with calmodulin resin (Roche), eluted using 5 mM EDTA, and protein containing fractions treated with lambda phosphatase (NEB) for 1 h at 4 °C, concentrated and loaded onto a Superdex 200 gel filtration column equilibrated in buffer DDK. Fractions containing DDK were pooled and dialyzed against buffer DDK containing 200 mM K-glutamate instead of 400 mM NaCl. Protein was then concentrated and snap frozen in liquid nitrogen for storage.

GST-Cdc6 was purified from BL21 DE3 Codon1 RIL bacterial cells (Stratagene) transformed with plasmid pAM1[40]. Expression was induced for 3 h at 25 °C at an OD$_{600}$ of 0.4–0.6. Cells were collected by centrifugation, resuspended in 50 mM K$_2$HPO$_4$/KH$_2$PO$_4$, pH 7.5, 5 mM MgCl$_2$, 1% Triton X-100 and 1 mM DTT (buffer A) + 2 mM ATP, 0.15 M KOAc and protease inhibitors (Roche), and 100 mg ml$^{-1}$ lysozyme added. After incubation for 30 min on ice, sample was sonicated and cleared by centrifugation at 20,000×g for 30 min. The supernatant was mixed with glutathione Sepharose resin (GE Healthcare) for 1 h at 4 °C. After washing with 20 column volumes of buffer A supplemented with 150 mM KOAc and 2 mM ATP, Cdc6 was cleaved off the beads in the same buffer supplemented with preScission protease (GE Healthcare) for 2 h at 4 °C. The flow-through was recovered, diluted such that the concentration of KOAc was reduced to 75 mM and incubated with hydroxyapatite resin washed in buffer A + 75 mM KOAc and 2 mM ATP. The column was washed with buffer A + 2 mM ATP, buffer A + 150 mM KOAc and 15% glycerol, and Cdc6 eluted with buffer A + 400 mM KOAc and 15% glycerol.

Protein containing fractions were pooled and snap frozen in liquid nitrogen for storage.

**Sample preparation of loaded double hexamers**. To immobilize a 2.8 kb DNA fragment containing ARS1, M280 streptavidin magnetic resin (Thermo Fischer Scientific) was washed twice with 10 mM Tris pH 7.2, 1 mM EDTA, 2 M NaCl (buffer 1) and resuspended with 50 ng 5′-biotinylated DNA per µl of starting slurry, and an equal volume of buffer 1. After mixing for 30 min at room temperature, resin was washed twice with 10 mM Hepes pH 7.6, 1 mM EDTA, 1 M K-acetate (buffer 2), twice with 10 mM Hepes pH 7.6, 1 mM EDTA (buffer 3), and resuspended in buffer 3 to a final volume half that of the starting slurry volume.

Each loading reaction was 80 µl and contained 25 mM Hepes pH 7.6, 10% glycerol, 10 mM Mg-acetate, 0.2% NP-40 substitute, 5 µl resin-immobilized DNA, 5 mM ATP, 7.5 nM ORC, 50 nM Cdc6, and 100 nM Cdt1–Mcm2–7. Reactions were incubated for 20 min at 30 °C before addition of DDK to 80 nM and further incubation for 30 min at 30 °C. Reactions were washed twice with 400 µl 25 mM Hepes 7.6, 5 mM Mg-acetate, 0.2% NP-40 substitute, 10% glycerol and 0.3 M KCl (high salt buffer), and once with 100 µl 25 mM Hepes 7.6, 5 mM Mg-acetate and 250 mM K-glutamate (EM buffer). For each cryo grid, 6 reactions were pooled, resuspended in 8 µl EM buffer supplemented with 0.4 U/µl MseI restriction enzyme (New England Biolabs) and incubated at 30 °C for 10 min with shaking. The supernatant was collected, moved to a fresh Eppendorf tube, and the supernatant collected a second time.

**Cryo-grid preparation and data collection**. A 4-µl sample of loaded double hexamers, either unmodified or DDK phosphorylated, were applied onto freshly glow discharged 400-mesh lacey grids containing an ultrathin layer of carbon (Agar Scientific). After a 2 min incubation the grid was double side blotted using a Vitrobot Mark IV (FEI ThermoFisher) for 0.5 s. To increase the concentration of particles, an additional 4-µl drop of sample was applied to the same grid and incubated for a further 2 min. The grid was double-side blotted for 2.0 s and plunged into liquid ethane. Cryo grids were screened for ice thickness and particle concentration using an FEI Spirit LaB6, operated at 120 kV and equipped with an Ultrascan 1000 charged couple device camera (Gatan, Inc). High-resolution data were collected on a Titan Krios electron microscope equipped with a K2 Summit direct electron detector (Gatan, Inc) at NeCEN, eBic (Diamond) and the Francis Crick Institute.

**Image processing**. Movies were corrected for beam induced motion using MotionCor2[41], where the first two frames were omitted and 5 × 5 patch alignment was used for the rest of the frames. All particles were semi-automatically picked using e2boxer from EMAN2[42] (version 2.07). CTF correction was performed using CTFFIND4[43] or Gctf[44] and the best quality integrated movies were inspected and selected in Relion-2[45]. Particles were extracted from dose-weighted micrographs and several rounds of 2D classification, 3D classification, and 3D refinement were performed in Relion-2 using particles binned by 2 (yielding a box size of 128 pixel) giving a pixel size ranging from 2.7 to 2.76 Å per pixel for all data sets. The yeast MCM double hexamer cryo-EM structure[24] (EMD-3663) was used as the initial model (filtered to 60 Å) for a cascade of 3D classification (five classes) and 3D refinement rounds. The best subset of particles were unbinned, yielding a 256-pixel box-size and a pixel size of 1.36–1.4 Å. Cryo-SPARC[46] was used for one round of ab-initio 3D classification (3 classes, C1 symmetry) and 3D refinement (with C2 symmetry imposition, refer to Supplementary Figs. 2 and 4 and Supplementary Table 1).

To retrieve DNA-bound particles, signal subtraction was performed in Relion-2. The ATPase rings were first isolated from the best refined structures using the segmentation tool in UCSF Chimera[47] and used to generate a mask in Relion-2. The re-projected masked density was used for signal subtraction at the level of aligned raw particles. As a result, a new particle data set was generated, which only contained the NTD and any density populating the ATPase pore of MCM. These signal-subtracted particles were subjected to 2D classification to allow identification and selection of DNA-bound particles. DNA-containing particles were reverted to their original density and used for ab-initio 3D classification and 3D refinement in cryoSPARC[46]. All final maps were post-processed in Relion-2 (Supplementary Table 1). The local resolution of the maps was estimated using ResMap[48].

**Atomic modeling and morphing**. A template model for duplex DNA was generated using make-na server (http://structure.usc.edu/make-na/server.html) and manually adjusted in Coot[49] to fit the electron density map. Phenix.geometry_minimization[50] was also employed to regularize the model geometry. The available atomic model of yeast MCM (PDB entry 3JA8) was manually adjusted in Coot[49] and subjected to real-space refinement using Phenix[50]. Morph movies and figures were generated using UCSF Chimera[47].

**Data availability**. Unmodified MCM double hexamer complex and DNA-bound structures have been deposited under EMDB entries EMD-3834 and EMD-3960, respectively. The DDK-phosphorylated MCM double hexamer and DNA-bound structures have been deposited under EMDB entries EMD-3833 and EMD-4164, respectively. Composite atomic model of the MCM double hexamer has been

deposited under accession code PDB 6F0L based on the unphosphorylated MCM double-hexamer cryo-EM maps.

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

## Acknowledgements

We would like to thank Julio Ortiz, Ludovic Renault (NeCEN) and Dan Clare, Alistair Siebert (eBIC) with assistance with cryo-EM data acquisition. Thanks to Raffaella Car-zaniga and Lucy Collinson (The Crick) for assistance with cryo-EM grid screening. Thanks to Andrew Purkiss and Phil Walker (The Crick) for help with computing. A.C. would like to acknowledge Peter Cherepanov and Peter Rosenthal for generosity and valuable discussion. This work was also supported by the Francis Crick Institute, which receives its core funding from Cancer Research UK (FC001065, FC001066), the UK Medical Research Council (FC001065, FC001066), and the Wellcome Trust (FC001065, FC001066). This work was also funded by a Wellcome Trust Senior Investigator Award (106252/Z/14/Z) and a European Research Council Advanced Grant (669424-CHRO-MOREP) to J.F.X.D. F.A.A. is the recipient of a PhD fellowship from the Boehringer Ingelheim Fonds.

## Author contributions

F.A.A., M.E.D., J.F.X.D, and A.C. designed the study. M.E.D. prepared the specimen supervised by J.F.X.D. F.A.A., J.L., A.N. prepared cryo-EM grids and acquired micro-graphs. F.A.A. performed all image processing and atomic modeling with the help of V.E.

P. A.C. supervised the project. F.A.A. and A.C. wrote the manuscript with inputs from M. E.D. and J.F.X.D.

## Additional information

**Competing interests:** The authors declare no competing financial interests.

