## [Peer Review File · Nature Communications]

Reviewers' comments:

Reviewer #1 (Remarks to the Author):

In this manuscript, the authors analyze by cryo-EM specifically the transition from pre-replication complex (pre-RC) to its activation. To this end, they determined the structure of the MCM-DNA complex and compared it with the structure of the CMG complex. The molecular mechanism involved in the helicase activation at replication origins is poorly known, and this study brings an important contribution to this field. In addition, the cryo-EM analysis is nicely performed, well explained, and provides helpful models to understand MCM movement on DNA. I am also not aware of pictures at such a resolution to allow discriminating the major and minor DNA grooves.

Minor point:

The authors provide interesting comments about the absence of melting of the DNA double helix at the pre-RC. However, the comparison with DnaA is not clear to me, as in *E. coli*, pre-RC formation includes the wrapping of DNA by DnaA and a high torsional strain that helps DNA unwinding at the origin.

Reviewer #2 (Remarks to the Author):

The manuscript by Abid Ali et al. describes cryo-EM structures of the eukaryotic MCM AAA+ helicase complex that is critical for licensing DNA replication and promoting DNA melting during initiation. The structures, at near atomic-resolution (4.3 and 4.8 angstroms), are of an unphosphorylated and DDK-phosphorylated double hexamer and considered to be in an "inactive" configuration prior to DNA melting which occurs following recruitment of Cdc45 and the formation of the CMG complex. Given the number of recent structures of MCM in different states of assembly (including, in particular, the MCM double hexamer by Li et al., 2015 and the CMG complex by Georgescu et al., 2017), a main achievement of this study is in resolving duplex DNA for structures at 7 and 7.5 angstroms, revealing a kinked path through the double-hexamer and interactions with the Watson-Crick strands along the length of the complex. Although the arrangement of the MCM double-hexamer structure seems to be identical to Li et al., which is unbound to DNA, the identification of the bound DNA and the use of single-particle masking and sorting methods in order to resolve DNA in their structures is an important advancement. From their higher resolution structures the authors also propose that the N-terminal tails of MCM4,6 become rigid upon phosphorylation, enabling Sld3,7 recruitment. Finally, by comparison to the structure of the CMG complex, they propose a model for DNA melting during DNA replication. While the structures are a significant advancement there is some concern over the data that support these models, which warrant consideration before publication. The concerns are the following:

-In Fig. 3 the data supporting that the NT tails of Mcm4,6 become rigid or "structured" is not convincing. Certainly, these tails do not become structured, as they state, given how poorly resolved the density is in Fig. 1c and f. In particular, the density identified in the +DDK 2D averages in 1c is very weak and barely identifiable. Difference images of the +/- DDK averages or variance images that show high variance in this region for the +DDK may help support their conclusion. Additionally, the purple density in the difference map in Fig 1f seems to be resolvable only at very low threshold values given the sigma of the difference map. Please include a map at the threshold where this density becomes apparent (preferably with the docked model) side by side with the -DDK map at the same threshold. Does Sld3,7 bind to this phosphorylated complex? Was this structure explored? Is there data to show that this sample was fully phosphorylated? Overall, it is difficult to conclude that this is bonafide protein density (not just additional flexibility) and results from the N-terminal tails of MCM4,6 (and not another protein such as Cdc7, which is

present in the silver stain). These statements should be dampened and additional supporting data is important.

-C2 symmetry was applied throughout. This seems reasonable given the apparent symmetry in the 2D averages. However, this constrains the position of the DNA potentially biasing its arrangement across the two hexamers in the DNA bound structures. If the symmetry is relaxed during the refinement is there any change in the DNA arrangement?

-The morph model to show conformational changes between the MCM DNA bound structure reported here and the CMG complex by Georgescu et al is compelling, however the CMG complex contains a single MCM hexamer and is bound to single stranded DNA compared to duplex DNA for the structures reported here. Thus, these structures are very different in composition, and therefore their conformations may not be on-path or depict the changes that occur during DNA melting.

Other minor concerns:

-The introduction should discuss recent structural achievements on MCM complexes and ORC complexes.

-Acronyms should be explained in the abstract and Introduction.

-Figure 1d – the docked model is difficult to see, the map should be more transparent or possibly grayscale with a colored model.

-Line 217-218: how can you concluded that hexamer sliding can still occur? There are extensive interactions along the length of the DNA, would these have to release cycle conformations in order to slide?

-Line 219-220: "each MCM in the double hexamer binds both DNA filaments" is misleading unless you can show that DNA present in both hexamers in an asymmetric reconstruction.

Reviewer #3 (Remarks to the Author):

Assembly of the CMG complex requires several steps, and their structural characterization is essential to understand DNA replication. In this manuscript, Abid Ali et al. improve our understanding of DNA replication by defining the structure of DNA-bound MCM, phosphorylated and not phosphorylated, using cryo-EM. The authors find that upon DDK phosphorylation MCM does not alter its conformation, except in a small region that would generate a binding site for Sld3/7. In addition, they describe a first model of the MCM-bound DNA. This structure is compared with the published structure of yeast CMG engaged with a replication fork. The work is technically well performed, although some aspects of the methodology are not explained sufficiently in the manuscript.

Issues:

- The title of the manuscript ("Cryo-EM structure of a licensed origin and the mechanism of DNA melting at the onset of eukaryotic DNA replication") seems overambitious. The main contribution of the work is solving the structure of the DDK-phosphorylated DNA-bound MCM complex, but the authors find that the conformation of the complex does not change significantly. They also describe the structure of MCM-bound DNA. These are interesting and needed observations, but they only provide limited new mechanistic insights.

- In the manuscript, it is unclear how the authors know if all MCMs are phosphorylated in their experimental conditions, and this could affect the interpretation of the structural analysis.

- The main finding for DDK-phosphorylated MCM is a new region assigned to the N-terminal tails of Mcm4 and 6, but then the authors do not elaborate on this finding. Some information that I miss:

1. The N-terminal tails of Mcm4 and 6 are only visible at low contoured levels. Have the authors checked if the density is present in all DDK-phosphorylated particles in their reconstruction? Can the authors classify the data based on the presence or absence of this density and provide numbers of how many particles show a clear density?
2. Similarly, does classification reveal localization heterogeneity or flexibility?
3. Can the authors estimate the local resolution of the new region found in DDK-phosphorylated MCM? The local resolution map in the supplemental information does not show this region, when this is the region the authors interpret from the map as a new finding.
4. Since the resolution of this region is probably poor, can it be ruled out that the new density found in the phosphorylated complex is unrelated with phosphorylation? Maybe this density, which can only be seen at low contour level, could also be found in unphosphorylated data if searched for it, or in a different data set as part of the flexibility of the region, regardless of phosphorylation.
5. Figure 1f. Legend mentions: "On the right: difference map for the phosphorylated Mcm4-6 N-terminal tails", but this is not found in the figure.
6. Figure 1f mentions: "To obtain the difference map the unmodified MCM was subtracted from the DDK phosphorylated MCM."
Is the difference found between both structures statistically significant? If not, then the difference of the density could not be unambiguously assigned to the effect of phosphorylation.

Minor comments

- If loading of two hexameric MCM helicases occurs only on the DNA, as indicated in the introduction, but most averages revealed no occupancy by the DNA, how are the double hexamers maintained after sliding?
- Can it be ruled out that the density in the phosphorylated complex corresponds to some DDK left? Trace amounts of DDK that had not been washed away could maybe be undetected biochemically, but still be part of a percentage of molecules.

Response to reviewers:

Reviewer #1:

We are pleased to learn that this reviewer deems our study ‘an important contribution to this field’, and that the cryo-EM analysis is considered ‘nicely performed, well explained, and provides helpful models’.

Minor point:

The authors provide interesting comments about the absence of melting of the DNA double helix at the pre-RC. However, the comparison with DnaA is not clear to me, as in *E. coli*, pre-RC formation includes the wrapping of DNA by DnaA and a high torsional strain that helps DNA unwinding at the origin.

We agree that the comparison with DnaA needs more explanation. We have revised the discussion and now recognize that, despite some similarities, the mechanisms of origin DNA opening in bacteria and eukarya differ fundamentally, in that no DNA wrapping is required for eukaryotic origins to be melted.

Reviewer #2 (Remarks to the Author):

We thank this reviewer for the useful comments. We are pleased that this reviewer considers our structures a ‘significant advancement’. In particular, we thank the reviewer for recognizing that “identification of the bound DNA and the use of single-particle masking and sorting methods in order to resolve DNA in their structures is an important advancement”.

1. In Fig. 3 the data supporting that the NT tails of Mcm4,6 become rigid or “structured” is not convincing. Certainly, these tails do not become structured, as they state, given how poorly resolved the density is in Fig. 1c and f.

This is an important point. One major concern of the reviewer is related to our phrasing. We think that an element in the tails likely becomes structured, although it remains flexibly tethered to the core of the MCM complex. We have modified the results section to clarify this point.

2. In particular, the density identified in the +DDK 2D averages in 1c is very weak and barely identifiable. Difference images of the +/- DDK averages or variance images that show high variance in this region for the +DDK may help support their conclusion.

We thank the reviewer for this useful idea. In Figure 1c (third row), we now show a difference image obtained by subtracting a representative –DDK 2D average from a + DDK 2D average. This exercise allows the identification of two symmetry-related features that decorate the N-terminal dimerization domain of MCM. We have also enlarged panel c in Figure 1 to better show this difference.

3. Additionally, the purple density in the difference map in Fig 1f seems to be resolvable only at very low threshold values given the sigma of the difference map. Please include a map at the threshold where this density becomes apparent (preferably with the docked model) side by side with the -DDK map at the same threshold.

We have added this information in Supplementary Figure 5.

4. Does Sld3,7 bind to this phosphorylated complex?

In EMBO J. 2016 May 2;35(9):961-73, Deegan and Diffley show that Sld3 binds to the MCM double hexamer in a DDK-dependent manner. We have now modified the introductory paragraph of the results section to clarify this point.

5. Was this structure explored?

A stable Sld3-7/phospho-MCM complex, tractable for structural studies, is difficult to obtain and will be the focus on a future study.

6. Is there data to show that this sample was fully phosphorylated?

Under the conditions used in our study, phosphorylation of the MCM complex by DDK takes place on at least 14 sites on Mcm6 and 25 sites on Mcm4 (Deegan et al, 2015), which are significantly redundant with one another. The SDS PAGE gel in Figure 1b shows that a high proportion of Mcm6 is shifted in response to DDK (now indicated by an arrow). Mcm4 also shifts, but it is hard to detect here, as it runs at the same position as Mcm2 and Mcm7. Unpublished experiments with radiolabelled ATP show that the non-shifted Mcm6 is also phosphorylated. Therefore, we expect that most loaded MCM is phosphorylated at a high proportion of DDK target sites. In line with this interpretation, the DDK concentration used here is saturating for both CMG assembly and DNA replication (two DDK-phosphorylation dependent events).

7. Overall, it is difficult to conclude that this is bonafide protein density (not just additional flexibility) and results from the N-terminal tails of MCM4,6 (and not another protein such as Cdc7, which is present in the silver stain).

We can resolve the same secondary structure elements in the non-phosphorylated and phosphorylated MCM structures. We therefore conclude that phosphorylation does not introduce flexibility in the core of the MCM structure. Conversely, we can rule out the possibility that the additional electron density feature observed in the phosphorylated sample is Cdc7, because in Figure 1b lane 2 (phosphorylated, DNA bound MCM) we fail to observe any trace of DDK (loaded as a control in lane 3). Our *preferred interpretation* of these data is that the additional density corresponds to the Mcm4/6 N-terminal tails, which gain some degree of rigidity upon DDK phosphorylation.

8. These statements should be dampened and additional supporting data is important.

As stated above, we recognize that our choice of words in describing the Mcm46 Nt tail should be improved and we now describe it as a structured region that likely remains flexibly tethered to the main core of the MCM complex (see changes in both Results and Discussion). Supporting data to substantiate our claims are now included in Supplementary Figures 5.

9. C2 symmetry was applied throughout. This seems reasonable given the apparent symmetry in the 2D averages. However, this constrains the position of the DNA potentially biasing its arrangement across the two hexamers in the DNA bound structures. If the symmetry is relaxed during the refinement is there any change in the DNA arrangement?

This is an important point. In Supplementary figure 10 we now include a structure of MCM-DNA refined in C1. The twisting of the DNA appears preserved in the C1 and C2 structures and the DNA-protein contacts remain unperturbed. We have modified the Results section to highlight this observation. We thank the reviewer for raising the issue of symmetrisation.

10. The morph model to show conformational changes between the MCM DNA bound structure reported here and the CMG complex by Georgescu et al is compelling, however the CMG complex contains a single MCM hexamer and is bound to single stranded DNA compared to duplex DNA for the structures reported here. Thus, these structures are very different in composition, and therefore their conformations may not be on-path or depict the changes that occur during DNA melting.

We thank this reviewer for raising an important point. We recognize that the transition between a double-hexameric MCM (bound to duplex DNA) to a single CMG (engaged to a DNA fork) might visit unknown DNA melting intermediates. Having taken on-board the reviewer's remark, we modified the molecular morph movie, removing the interpolations of the DNA structure, upon transition from MCM-DNA to CMG-DNA. We have modified the text to acknowledge the potential caveats of our model.

Other minor concerns:

12. The introduction should discuss recent structural achievements on MCM complexes and ORC complexes.

Done.

13. Acronyms should be explained in the abstract and Introduction.

Done. Thanks for pointing this out.

14. Figure 1d – the docked model is difficult to see, the map should be more transparent or possibly grayscale with a colored model.

Done. Thanks.

15. Line 217-218: how can you concluded that hexamer sliding can still occur?

There are extensive interactions along the length of the DNA, would these have to release cycle conformations in order to slide?

We fully agree with this reviewer. We cannot rule out from our data that a short-lived intermediate exists where MCM pore loop elements let go of the DNA and this is the state competent for sliding. Such a short-lived molecular specie would be poorly populated and not detectable by 3D classification methods. We have now modified our discussion to entertain this possibility.

16. Line 219-220: "each MCM in the double hexamer binds both DNA filaments" is misleading unless you can show that DNA present in both hexamers in an asymmetric reconstruction.

We now include asymmetric reconstructions showing pore loop contacts for both Watson and Crick strands in both rings, supporting our claim (Supplementary Figure 10 and text changes in the Results section).

Reviewer #3 (Remarks to the Author):

We thank this reviewer for stating that with our study we ‘improve our understanding of DNA replication’ and that our ‘work is technically well performed’. Below is a point-by-point reply to the issues raised.

1. The title of the manuscript (“Cryo-EM structure of a licensed origin and the mechanism of DNA melting at the onset of eukaryotic DNA replication”) seems overambitious. The main contribution of the work is solving the structure of the DDK-phosphorylated DNA-bound MCM complex, but the authors find that the conformation of the complex does not change significantly. They also describe the structure of MCM-bound DNA. These are interesting and needed observations, but they only provide limited new mechanistic insights.

We thank this reviewer for stating that our observations are ‘interesting and needed’. Following this reviewer’s suggestion, we have changed our title to “Cryo-EM Structure of a Licensed DNA replication origin”.

2. In the manuscript, it is unclear how the authors know if all MCMs are phosphorylated in their experimental conditions, and this could affect the interpretation of the structural analysis.

Thanks for raising an important point, which we address in our reply to Referee #2 point 6.

3. The main finding for DDK-phosphorylated MCM is a new region assigned to the N-terminal tails of Mcm4 and 6, but then the authors do not elaborate on this finding. Some information that I miss: The N-terminal tails of Mcm4 and 6 are only visible at low contoured levels. Have the authors checked if the density is present in all DDK-phosphorylated particles in their reconstruction? Can the authors classify the data based on the presence or absence of this density and provide numbers of how many particles show a clear density?

Prompted by the reviewer’s comment, we have further analysed the +DDK MCM structure by local 3D classification. To this end we have generated a three-dimensional mask containing the phospho-dependent N-terminal feature and analysed it by 3D classification without alignment. In

Supplementary Figure 5, we show 2D classes (no symmetry), with phospho-dependent N-terminal elements decorating both N-terminal domains in the MCM double hexamer.

3D classification failed to identify a subset of particles that clearly lack the new density feature. We have modified the Results section to explain this analysis.

4. Similarly, does classification reveal localization heterogeneity or flexibility?

The local 3D classification described above revealed two distinct configurations for the phospho-Nt element, falling into two main categories, namely extended and compacted (shown in two refined structures in Supplementary Figure 5a). Two-dimensional classification of particles contributing to the two structures confirms our observation. We have modified the Results section to describe these data.

5. Can the authors estimate the local resolution of the new region found in DDK-phosphorylated MCM? The local resolution map in the supplemental information does not show this region, when this is the region the authors interpret from the map as a new finding.

As requested, in Supplementary Figure 5b,c, we now include a local resolution map of phospho-MCM (+DDK), displayed at a threshold where Mcm46 Nt elements are visible.

6. Since the resolution of this region is probably poor, can it be ruled out that the new density found in the phosphorylated complex is unrelated with phosphorylation? Maybe this density, which can only be seen at low contour level, could also be found in unphosphorylated data if searched for it, or in a different data set as part of the flexibility of the region, regardless of phosphorylation.

To address this question, we have taken two approaches. *i.* We compare the resolution map of non-phosphorylated and phosphorylated MCM, shown at matching contour level. *ii.* We have attempted to analyse -DDK MCM using a local 3D classification and imposing the same local mask employed for the MCM +DDK analysis. We failed to identify any density surrounding the Mcm4/6 Nterminal region in the -DDK MCM dataset.

7. Figure 1f. Legend mentions: "On the right: difference map for the phosphorylated Mcm4-6 N-

terminal tails”, but this is not found in the figure.

We have now modified the figure legend to better explain Figure 1f. The purple density is the phosphorylated [minus] non-phosphorylated MCM difference map shown at 0.08 sigma. More information on how this difference map was generated is now contained in Supplementary Figure 5a.

8. Figure 1f mentions: “To obtain the difference map the unmodified MCM was subtracted from the DDK phosphorylated MCM.”

Is the difference found between both structures statistically significant? If not, then the difference of the density could not be unambiguously assigned to the effect of phosphorylation.

The local classification analysis suggested by this referee, and described in points 3.-7., enhances our confidence in the phospho-dependent N-terminal MCM element that departs from Mcm4/6. 3D classification allowed us to identify two states for the phospho-Nt tails, independently detected in 2D averages and refined 3D structures. We feel the reviewer’s suggestion has significantly improved our study.

Minor comments

9. If loading of two hexameric MCM helicases occurs only on the DNA, as indicated in the introduction, but most averages revealed no occupancy by the DNA, how are the double hexamers maintained after sliding?

MCM double hexamer formation depends on DNA replication initiators ORC and Cdc6, which recognize specific sequence motifs at DNA replication start sites. DNA is therefore required for MCM double hexamer formation because it recruits the MCM-loading function (i.e. double hexamer formation function), provided by ORC/Cdc6. Li et al (Nature 2015 paper) report a structure of the DNA-free endogenous MCM double hexamer, isolated after enzymatically digesting DNA. A DNA-free stable double-hexameric MCM had therefore already been reported. We have modified the results section to clarify this point.

10. Can it be ruled out that the density in the phosphorylated complex corresponds to some DDK left? Trace amounts of DDK that had not been washed away could maybe be undetected biochemically, but still be part of a percentage of molecules.

We do not think that this is a real possibility, also see reply to Reviewer #2, point 7. By inspecting the 2D averages and the Resmap included in Supplementary Figure 5b-c, it appears evident that the phospho Mcm4/6 Nt element is as poorly resolved as the winged helix domain appendices that decorate the C-terminal face of an MCM hexamer. These flexible C-terminal elements become

visible in the activated CMG. Likewise, the flexible phospho Mcm4/6 Nt elements might become more rigid during subsequent stages of replisome maturation (e.g. Sld3/7/Cdc45 recruitment).

REVIEWERS' COMMENTS:

Reviewer #1 (Remarks to the Author):

The authors have satisfactorily answered to my minor comments. This is a nice and stimulating piece of work. I also found that they correctly answered to the technical comments raised by the other reviewers obviously more familiar with cryo-EM.

Reviewer #2 (Remarks to the Author):

The authors have adequately responded to all the reviewers comments and included new data that help support their conclusions about the location of the MCM tails and organization of DNA across the two rings. I support publication of this manuscript.

Reviewer #3 (Remarks to the Author):

The new experiments performed by the authors have improved the characterization of the new density that appears in DDK-phosphorylated MCM. After review, it is now convincing that this density appears after phosphorylation.

Minor comments:

- It is still not proved that this new density corresponds to the N-terminal tails of Mcm4 and Mcm6.

As the authors mention in the rebuttal letter, their preferred interpretation is that this density corresponds to the N-terminal tails of Mcm4 and 6, and consequently a landing platform for Cdc45.

Although I agree with the authors, I also agree that the results presented do not 100 % prove this assignment, and I would modify the text in two places:

----- Lines 113-118. "Given these results, we assign the new flexible elements to the N-terminal tails of Mcm4 and Mcm6...."

I would say something like "Given these results, our preferred interpretation is that these new flexible elements can be assigned to the N-terminal tails of Mcm4 and Mcm6...."

----- Abstract, lines 21-22. "...and visualise a phospho-dependent MCM element important for Cdc45 recruitment"

I would rather say "...and visualise a phospho-dependent MCM element we propose to be important for Cdc45 recruitment"

Response to reviewers:

Reviewer #3:

We thank this reviewer for the positive assessment of our revised manuscript. Below is a point-by-point reply to the referee's comments.

- It is still not proved that this new density corresponds to the N-terminal tails of Mcm4 and Mcm6. As the authors mention in the rebuttal letter, their preferred interpretation is that this density corresponds to the N-terminal tails of Mcm4 and 6, and consequently a landing platform for Cdc45. Although I agree with the authors, I also agree that the results presented do not 100 % prove this assignment, and I would modify the text in two places:

----- Lines 113-118. "Given these results, we assign the new flexible elements to the N-terminal tails of Mcm4 and Mcm6...."

I would say something like "Given these results, our preferred interpretation is that these new flexible elements can be assigned to the N-terminal tails of Mcm4 and Mcm6...."

We have edited the sentence which now reads:

"Given these results, our preferred interpretation is that new flexible element corresponds to the N-terminal tails of Mcm4 and 6, which become visible upon DDK phosphorylation."

----- Abstract, lines 21-22. "...and visualise a phospho-dependent MCM element important for Cdc45 recruitment"

I would rather say "...and visualise a phospho-dependent MCM element we propose to be important for Cdc45 recruitment"

We have edited this sentence in the abstract. It now reads:

To understand origin activation, here we have determined the cryo-EM structures of DNA-bound MCM, either unmodified or phosphorylated, and visualise a phospho-dependent MCM element likely important for Cdc45 recruitment.